# A Patient Selection Approach Based on NTCP Models and DVH Parameters for Definitive Proton Therapy in Locally Advanced Sinonasal Cancer Patients

**DOI:** 10.3390/cancers14112678

**Published:** 2022-05-28

**Authors:** Alfredo Mirandola, Stefania Russo, Maria Bonora, Barbara Vischioni, Anna Maria Camarda, Rossana Ingargiola, Silvia Molinelli, Sara Ronchi, Eleonora Rossi, Alessandro Vai, Nicola Alessandro Iacovelli, Juliette Thariat, Mario Ciocca, Ester Orlandi

**Affiliations:** 1Medical Physics Unit, Clinical Department, CNAO National Center for Oncological Hadrontherapy, 27100 Pavia, Italy; stefania.russo@cnao.it (S.R.); silvia.molinelli@cnao.it (S.M.); eleonora.rossi@cnao.it (E.R.); alessandro.vai@cnao.it (A.V.); mario.ciocca@cnao.it (M.C.); 2Radiotherapy Unit, Clinical Department, CNAO National Center for Oncological Hadrontherapy, 27100 Pavia, Italy; maria.bonora@cnao.it (M.B.); barbara.vischioni@cnao.it (B.V.); annamaria.camarda@cnao.it (A.M.C.); rossana.ingargiola@cnao.it (R.I.); sara.ronchi@cnao.it (S.R.); ester.orlandi@cnao.it (E.O.); 3Department of Radiation Oncology, Fondazione IRCCS Istituto Nazionale dei Tumori, 20133 Milano, Italy; nicolaalessandro.iacovelli@istitutotumori.mi.it; 4Department of Radiation Oncology, Françoise Baclesse Center ARCHADE, Normandy University, 14000 Caen, France; jthariat@gmail.com

**Keywords:** locally advanced sinonasal cancer, proton therapy, photon/proton plan comparison, NTCP, model-based selection

## Abstract

**Simple Summary:**

The role of proton therapy as a radiation treatment option for locally advanced sinonasal cancer patients has increased in the last years, showing promising results in terms of clinical outcomes. Definition of strategies to identify patients who would benefit the most from proton therapy in terms of reduced toxicity is highly desirable, due to limited availability and higher costs of this treatment option. The novelty of our in silico study relies on assessing the suitability of a mixed dose volume histograms and normal tissue complication probability model-based approach, aiming at providing, to the scientific community, a patient selection criteria possibly leading the therapeutic choice between proton therapy and advanced photon techniques.

**Abstract:**

(1) Background: In this work, we aim to provide selection criteria based on normal tissue complication probability (NTCP) models and additional explanatory dose-volume histogram parameters suitable for identifying locally advanced sinonasal cancer patients with orbital invasion benefitting from proton therapy. (2) Methods: Twenty-two patients were enrolled, and two advanced radiation techniques were compared: intensity modulated proton therapy (IMPT) and photon volumetric modulated arc therapy (VMAT). Plans were optimized with a simultaneous integrated boost modality: 70 and 56 Gy(RBE) in 35 fractions were prescribed to the high risk/low risk CTV. Several endpoints were investigated, classified for their severity and used as discriminating paradigms. In particular, when NTCP models were already available, a first selection criterion based on the delta-NTCP was adopted. Additionally, an overall analysis in terms of DVH parameters was performed. Furthermore, a second selection criterion based on a weighted sum of the ΔNTCP and ΔDVH was adopted. (3) Results: Four patients out of 22 (18.2%) were suitable for IMPT due to ΔNTCP > 3% for at least one severe toxicity, 4 (18.2%) due to ΔNTCP > 20% for at least three concurrent intermediate toxicities and 16 (72.7%) due to the mixed sum of ΔNTCP and ΔDVH criterion. Since, for some cases, both criteria were contemporary fulfilled, globally 17/22 patients (77.3%) would benefit from IMPT. (4) Conclusions: For this rare clinical scenario, the use of a strategy including DVH parameters and NTCPs when comparing VMAT and IMPT is feasible. We showed that patients affected by sinonasal cancer could profit from IMPT compared to VMAT in terms of optical and central nervous system organs at risk sparing.

## 1. Introduction

Proton radiotherapy (PT) represents a new paradigm for treating Paranasal Sinuses Cancers (PNSCs) [1]. Compared to photon-based Intensity Modulated Radiation Therapy (IMRT) and Volumetric Modulated Arc Therapy (VMAT), Intensity Modulated Proton Therapy (IMPT) has physical advantages arising from the inverse depth dose profile and a rapid dose fall-off that spares the healthy tissue distal to the tumor [2], leading to a high quality dose distributions. On the other hand, robust plan optimization is crucial to properly deal with in-patient particle range uncertanties. The IMPT delivered with pencil beam scanning (PBS) technique allows to scan or ’paint’ the tumor volume voxel-by-voxel and layer-by-layer, so delivering high doses to the targets while sparing the surrounding healthy tissues. Some recent papers including patients most frequently affected by PNSCs [Liang2018], olfactory neuroblastoma (ONB) [3], and undifferentiated sinonasal carcinoma (SNUC), reported promising outcome results and a low rate of late neurological toxicity [4,5,6,7], although proving a deep focus on overall ocular sequelae. Because of the low number of PT facilities available and the relatively higher costs of this treatment compared to IMRT, PT should be reserved for patients that are likely to benefit the most in terms of toxicity risk reduction. Recently, model-based clinical evaluations have been proposed as valid evidence-based methods alternative to randomized controlled trials [8,9]. The dose reduction to relevant organs at risk (OARs) resulting from plan comparison between proton and photon techniques is translated into a clinically relevant benefit, estimated in terms of reduced risk of side effects by means of normal tissue complication probability (NTCP) models. Patients are so qualified to receive PT if the difference in the predicted risks between the photon and the proton plan is larger than a defined threshold, e.g., 10% for a Grade 2 toxicity, which represents the minimal potential benefit to qualify the patient for PT [10]. Modelled side effects of radiotherapy (RT) in orbital, sinonasal, and skull-based districts have been developed including ocular toxicity [11], visual impairment [12,13,14], radiation necrosis [12,15,16] and cognitive deterioration [17,18]. However, for most of the above-mentioned side effects, only photon-derived NTCP models are available, often without external validation. Moreover, dose distributions obtained throught different radiation delivery techniques or, even more, different adopted particle types, may also affect the predictive power of NTCP models. Therefore, NTCP models developed for photons should always be validated with protons, prior to the direct comparison of toxicities rates [10]. Nevertheless, we think that the developement and application of models that are yet to be trained and validated for both photons and protons for the RT planning for locally advanced (LA) sinonasal cancers (SNCs) can help the clinician in choosing the best treatment RT approach. This retrospective in silico study was aimed to quantify the impact of using protons to treat 22 cases of LA or inoperable PNSCs, compared to VMAT. We performed a plan comparison analysis in terms of both NTCP models and dose-volume histograms (DVHs). In addition, both the expected specific RT-related and composite toxicities were analyzed, particularly for effect on ocular and neurological OARs.

## 2. Materials and Methods

### 2.1. Patients’ Characteristics

Twenty-two patients with LA or unresectable PNSCs, staged III-IVB and already treated between 2013 and 2021 at the National Center for Oncological Hadrontherapy (CNAO) in Italy, were selected for this retrospective planning investigation. Patient cohort included 15 SNUC, 6 ONB and 1 patient with NUT-midline carcinoma (NMC). All patients received definitive IMPT or mixed beam approach including boost of carbon ion and IMPT with a total dose of 66–74 Gy(RBE, Relative Biological Effectiveness weighted dose) with or without histology-driven induction and/or concomitant Chemotherapy (CHT). There were 11 female and 11 male patients whose median age was 57.5 years (range 23–81 years). Of note, between 2013 and 2018, a multicenter single-arm phase II clinical trial on LA inoperable SNCs, assessing the activity and safety of an innovative integration of multi-modality treatment, including induction CHT, photon, and charged particle RT, was ongoing. However, patients of the present analysis were not included in that study. All patients were enrolled in an institutional clinical registry, (CNAO REgistry triAL—REGAL, registered at ClinicalTrials.gov NCT05203250). Ethical approval by the institutional review board was obtained for this study (Protocol number. 20210047647). The patients signed an infromed consensus and agreed to use their anonymized data for scientific purposes.

### 2.2. Volumes Definition, Dose Prescription, and Planning Objectives

For all the patients, an expert radiation oncologist delineated two clinical target volumes (CTVs) according to previously reported definitions [19,20]. A high-risk CTV (HR-CTV) including the gross tumor volume (GTV) of the primary tumor was determined by clinical information, endoscopic procedure, and magnetic resonance imaging (MRI), before any CHT, with a margin in agreement with the compartment-related definition by Claus et al. [21]. A low-risk CTV (LR-CTV) was defined, including bilateral nodal levels Ib–III and retropharyngeal nodes irradiated electively, in compliance with the international guidelines [22]. Planning target volumes (HR-PTVs and LR-PTVs, respectively) were generated by adding a 3 mm margin to the corresponding CTVs. Although in clinical practice different dose prescriptions and fractionations schemes could be used, based on the histology, disease stage, extension, and response to induction chemotherapy, we used the same dose prescription scheme in order to better compare VMAT and IMPT for the study purpose. Thus, all plans were re-optimized with a simultaneous integrated boost (SIB) modality, with a prescribed dose of 70 Gy(RBE) and 56 Gy(RBE) in 35 fractions, to HR-CTV and LR-CTV, respectively. For IMPT plans, we used a fixed RBE value of 1.1. The contoured OARs included optic chiasm, optic nerves, retinae, anterior chambers, eyeballs, lacrimal glands, brainstem, spinal cord, temporal lobes, cochleae, and lenses. Parotid glands, mandible, and glottic larynx were also considered for patients needing elective or curative neck irradiation, but were not included in the present analysis. All optic structures were classified into ipsi-lateral and contra-lateral, according to primary tumor proximity. When tumor localization was central and symmetric respect to paired OARs, the structure receiving the higher mean dose has been considered as ipsi-lateral. In absence of clearly established MRI images, visual field tests analysis helped to determine the more impaired, potentially expendable side of the ocular structures. Structures located within the CTVs were not contoured because either they were absent due to resection or it was assumed that symptoms would be tumor-related and not therapy-related. Thus, we did not expect any outcome improvement from PT. In order to reduce inter-observer bias, an additional expert radiation oncologist checked for contoured OARs plausibility. For the investigation purpose, we have included the updated evidence in terms of suggested radiation dose-volume constraints for a variety of normal tissue complications related to head and neck cancer treatments, published after the QUANTEC reports (early 2010) and including Patient Reported Outcome measures (PROs), where available. The plan optimization process aimed to increase as much as possible the CTVs coverage without exceeding the constraints to selected OARs. In particular, we opted for a priority order for planning objectives and constraints. We gave the highest planning priority to the following structures: brainstem, spinal cord, optic chiasm, and contra-lateral optic nerve at least, to preserve mono-lateral vision. CTVs coverage represented a second priority and the lowest priority was given to the remaining OARs sparing. We considered as clinically acceptable and potentially deliverable to the patient all the plans passing the criteria summarized in Table 1, column 2.

### 2.3. Plan Optimization

For comparison, both proton and photon plans were optimized with Raystation (Raysearch laboratories AB, Stockholm, Sweden) Treatment Planning System (TPS) V8B version, available at CNAO at the time of this study. The VMAT plans were optimized adopting two co-planar 6MV X-Ray photon beam arcs, the first one with a clockwise gantry rotation from 185° to 175° with collimator angle of 20°, the second one with a counter clockwise gantry rotation from 175° to 185° with a collimator rotation of 340°. The dose distribution was determined employing the collapsed cone convolution algorithm. We included HR-PTV and LR-PTV in the optimization to ensure adequate CTVs coverage. For Proton plans, we used IMPT technique with PBS modality with the proton beam settings routinely adopted at CNAO in clinical practice [23]. Since no gantry is available at CNAO, we employed three horizontal fixed beams, two lateral and one vertex field, which represent the clinical beam configuration for sinonasal cancer patients. In order to give the full dose to the superficial parts of the CTVs, we included a 3.0 cm thick range shifter for optimization, placed with 2–3 cm air gap from the patient external contour. The RayStation Monte Carlo dose engine for dose calculation was employed. Differently from photons plans, PTVs were not introduced in the optimization. In order to achieve the CTVs coverage goals, we applied instead a robust planning strategy based on minimax optimization [24,25], considering both setup (±2 mm) and beam range (±3%) uncertainties. Both VMAT and IMPT plans were calculated with a 2 mm dose-calculation grid.

### 2.4. Plan Analysis and Comparison

For each patient case, we performed a DVH analysis similarly to the quoted references [19,26]. For target coverage evaluation, we focused only on the CTVs, since the PTVs were not included in the IMPT optimization. Regarding OARs sparing, we considered a combination of several dose parameters possibly associated to ocular and neurological toxicities, based on the literature as shown in Table 1, column 3. In particular, the volume of retina receiving doses higher than 50 Gy(RBE) (V${}_{50}$) and 55 Gy(RBE) (V${}_{55}$) was evaluated as potentially related to radiation retinopathy, according to [27,28].The volume of optic nerves and chiasm receiving doses higher than 55 Gy(RBE) was also estimated, being related to the risk of radiation-induced optic neuropathy (RION), as reported in QUANTEC series [14,18,29,30]. The dose to the lacrimal glands was also investigated and we considered both the volume receiving 30 Gy(RBE) (V${}_{30}$) and the mean dose, being possibly related to dry eye syndrome (DES) [31] as described in [32,33]. Finally, the volume of the brain receiving doses higher than 25 Gy(RBE) (V${}_{25}$) and 35 Gy(RBE) (V${}_{35}$) was also assessed, as potentially leading to fatigue or memory impairment [34]. For the dosimetric comparison we calculated the relative percentage difference for all the DVH parameters reported in Table 1 (column 3) between photon and proton plans. (DVH${}_{ph}$ and DVH${}_{p}$ respectively) as ΔDVH${}_{ph-p}$ = 100*[(DVH${}_{ph}$− DVH${}_{p}$)/DVH${}_{ph}$]. Finally, we also computed the Homogeneity Index (HI) = (D${}_{2\%}$− D${}_{98\%}$)/D${}_{pres}$, where D${}_{pres}$ is the prescription dose, and the Conformity Index (CI) = TVD${}_{95\%}$/TV, where TVD${}_{95\%}$ and TV represented the total volume encompassed by the D${}_{95\%}$ and the target volume respectively. Since PTVs were not used in the IMPT plans, the CTVs were used as TV. We applied the Wilcoxon signed-rank test with a p-value of 0.05 for testing the null hypothesis that a certain DVH parameter is equal for both the two sets of 22 VMAT and IMPT plans. Therefore, in the plan comparison analysis only the DVH indices with a statistically significant difference between the two samples were included. Afterwards, we converted ΔDVH${}_{ph-p}$ to an arbitrary variable (DVH*) which can assume three discrete values (−1, 0, 1), according to the following criteria: if ΔDVH${}_{ph-p}$ was higher than +20%, revealing an evident advantage from IMPT, then DVH* = +1; if ΔDVH${}_{ph-p}$ was lower than −20%, indicating a distinct benefit from VMAT, then DVH* = −1. In all other cases, DVH* = 0, meaning that the two radiation techniques are comparable for that specific OAR under investigation.

After a comprehensive review of the literature, eight NTCP models were used for plan comparison in this study (Table 2). We based the model selection on a focus on SNC-specific toxicities relying on clinical experience and considering studies developing, validating, or applying NTCP models with available parameters for evaluation. In this work the clinical toxicities endpoints were divided into two categories, intermediate and severe, depending on their impact on patients Quality of Life (QoL), as detailed in Table 2, in brackets. Late neurological toxicity with devastating clinical consequences or potentially life-threatening, such as blindness [12], brain, brainstem and spinal cord necrosis [15], temporal lobe injury [35], were defined as severe. Otherwise, other relevant rare adverse effects, which still have a significant but less tremendous impact on patients QoL, were referred as intermediate. We established the acute overall ocular toxicity ≥ Grade 2 as intermediate, according to the radiation toxicity criteria of Radiation Therapy Oncology Group (RTOG) and European Organization for Research and Treatment of Cancer (EORTC) as reported by Batth et al. [11], since the authors contemplated a wide spectrum of toxicity with variable impact on patients’ Qol, including conjunctivitis, keratitis and corneal ulceration. In contrast, DES was scored as a severe toxicity [36] since it is related to acute radiation reactions that ultimately resulted in compromised vision according to RTOG Grade 3 and 4 toxicities and National Cancer Institute’s Common Terminology for Adverse Advents (NCI CTCAE) for DES Grade 2 and 3. Subsequently, we defined late brain necrosis as intermediate [16] sinche the authors chose brain necrosis CTCAE v4.0 ≥ Grade 2 endpoint derived from MRI and clinical symptoms for their study. The net difference in NTCP for specific endpoint (Table 2) between photon and proton plans (phNTCP and pNTCP respectively) was calculated as ΔNTCP = ${}_{ph}$NTCP – ${}_{p}$NTCP, for severe and intermediate toxicities (ΔNTCP${}^{s}$, ΔNTCP${}^{i}$, respectively) and used for further analysis. We thus introduced a supplementary selection criterion for plan comparison as a mixed ΔNTCP-ΔDVH parameter called total score (TS), determined as reported in Equation (Equation 1). TS consisted in a weighted sum that considers the 8 NTCP models described in Table 2, four models for severe and four models for intermediate toxicities (ΔNTCP${}^{s}$ andΔNTCP${}^{i}$, respectively) were adopted, together with ΔDVH for the m DVH parameters that, according to Wilcoxon test, were statistically significant in terms of the percentage difference between IMPT and VMAT plans.
(1)$$TS=w_{1}\sum_{j=0}^{4}\Delta NTCP_{j}^{s}+w_{2}\sum_{k=0}^{4}\Delta NTCP_{k}^{i}+w_{3}\sum_{r=0}^{m}DVH_{r}^{*},$$

A weighting factor multiplies each term in Equation (Equation 1). We applied a relative weight of 20 to the ΔNTCP${}^{s}$ (w${}_{1}$), while we assigned a unitary weight factor for ΔNTCP${}^{i}$ (w${}_{1}$), given the impact on the patient QoL. Finally we assigned to w${}_{3}$ a value of 10, being both severe and intermediate toxicities considered in the third term of the equation. If at least one of the following two conditions was met, we expected the selected patient case to benefit from IMPT in terms of reduced risk of radiation-induced side effects:1(a) ΔNTCP${}^{i}$ exceeded a threshold of 20% (similar to [10]) for at least three of all the investigated intermediate toxicities side effects.(b) ΔNTCP${}^{s}$ exceeded a threshold of 3% for a single severe toxicity.2TS was higher than a certain arbitrary threshold of 250.

## 3. Results

### 3.1. Dosimetric Analysis

Median GTV, HR-CTV and LR-CTV volumes were 39.1 cc (range 17–107.5 cc), 135.1 cc (range 32.8–343.4 cc) and 441.7 cc (range 131.4–677.2 cc), respectively. A representative dose distribution for IMPT and VMAT is showed in Figure 1. All the DVH indices reported in Table 1 column 3 were statistically different between IMPT and VMAT, according to the Wilcoxon test, thus m in Equation (Equation 1) was equal to 14.

The CTVs coverage comparison results is summarized in Table 3. Although all plans were clinically acceptable, VMAT provided a slightly better target coverage than IMPT in terms of D${}_{95\%}$, D${}_{98\%}$, V${}_{95\%}$, with difference being within 1.5% considering the mean over all patients. Moreover, CI was lower for IMPT for both targets while HI was lower for photon plans. The results for the dose parameters associated to ocular and neurological toxicities are summarized in Table 4. Each cell contains the number of patient cases as a function of ΔDVH values: ΔDVH = +1 when IMPT performs better than VMAT, ΔDVH = −1 if VMAT performs better than IMPT, ΔDVH = 0 when no net benefit was found in either the two techniques. For brain, temporal and frontal lobes, IMPT was far superior to VMAT in terms of V${}_{25}$ and V${}_{35}$ for the entire patient cohort.

### 3.2. NTCP Analysis

Table 5 illustrates results for the investigated severe toxicities. We found an overall benefit from IMPT versus VMAT since ΔNTCP${}^{s}$ values were ≥0.01% for all the patient cases. Considering the brain necrosis NTCP model, both ${}_{ph}$NTCP and ${}_{p}$NTCP resulted to be less than 0.05%, thus showing that this toxicity would be hardly found for the investigated dose distributions, being the related toxicity dose constraints well fulfilled. The condition 1(b) was never matched for the blindness NTCP model [12] with meanΔNTCP${}^{s}$ being lower than 1% for optic structures. Regarding DES, ΔNTCP${}^{s}$ > 3% was found only in one case out of 22 for the contralateral lacrimal gland. 1(b) condition for neurological toxicity was met in 4 patient cases. As showed in Table 6, each specific ΔNTCP${}^{i}$ for intermediate toxicities and the mean values indicated that IMPT better spares the OARs resulting in a lower complication probability. VMAT was found to be superior for lens and lacrimal glands only in 26% of patient cases.

### 3.3. TS Calculation and Overall Results

Table 7 summarizes the results over the entire patient cohort, considering the selection criteria adopted to drive the choice between the two radiation techniques (1(a), 1(b) and 2). Values written in bold character show the number of patients for which IMPT plans were superior to VMAT plans, according to the investigated toxicities. In our investigation, 17 patients out of 22 (77.3%) would definitely benefit from IMPT considering the above-mentioned criteria. The condition 1 was fulfilled in 8 patients out of 22 (34.4%), specifically in 4 cases the 1(a) condition and in 4 cases the 1(b) condition. Condition 2 was fulfilled in 16 patients (72.7%), in 6 of which (27.3%) the 1(a)–1(b) and TS criteria were satisfied at the same time. For 5 patients no evidence of superiority of VMAT over IMPT has emerged.

## 4. Discussion

Unlike other most common head and neck cancer sites, there are few comparison studies concerning PNSCs. Some studies reported interesting findings when comparing protons versus photons in SNCs. As a general remark, PT allows to deliver lower doses to several OARs in PNSCs without significant difference in terms of target coverage, conformity or homogeneity index [39], with the highest benefit for ethmoid tumors [40]. The horseshoe-shaped target volumes and the proximity or involvement of several critical structures such as temporal lobe, brain, middle cranial fossa, clivus, orbital apex and orbit, have led high expertise radiation oncology community to administer PT in clinical practice, when available, even without an overall plan comparison study. Although many retrospective and prospective series have been published, long-term outcome and ocular details are quite limited [1]. However, these reports often included heterogeneous series regarding the number of patients, tumor histology and stages, PT techniques and settings, and therefore not distinguishing among definitive, postoperative or re-irradiation scenarios [1]. Moreover, the toxicity outcomes for definitive treatment requiring high prescribed dose levels, as those planned in this study, are rarely described. Zenda et al. [41] reported severe late toxicities including central nervous disorders and visual loss in 5 out of 39 (12.8%) patients with unresectable T4 SNCs treated with accelerated definitive PT with or without induction CHT, thus showing an overall safety profile considering the advanced stage disease. Another work by Toyomasu et al. [42] on LA and unresectable PNSCs treated with protons or carbon ions alone, showed severe toxicity in 7 out of 38 (18,4%) patients treated with PT at the total dose of 65–70.2 Gy(RBE) in 26 fractions. The reported toxicities were glaucoma, brain necrosis, retinopathy and optic nerve disorders in patients undergoing radical treatment and affected by T4 disease with tumors near the optic nerve, eyeball and brain. These clinical outcomes are quite expected in case of T4 diseases with tumors near the optic nerve, eyeball or brain, treated with a high dose in a radical treatment course. Conversely, in case of postoperative treatment with lower prescription doses, a reduced rate of severe side effects was showed [43]. Moreover, in a recent study conducted on a patient cohort treated with PBS technique [44] a comprehensive logistic regression model was proposed. It takes into account both patient specific clinical parameters such as age, tumor involvement, hypertension and gender and dosimetry for the onset of RION.

In this context of inhomogeneous investigations, some selection criteria based on reliable models and dosimetric parameters are undoubtedly needed. The novelty of our in silico study relies on assessing the suitability of a mixed DVH-NTCP model-based approach for patients with PNSCs suitable for definitive radiotherapy, aiming at providing a patient selection criteria possibly leading the therapeutic choice between IMPT and VMAT techniques. In our investigation, VMAT was assumed as the best enough photon technique, widely available worldwide. Nevertheless, alternative treatment techinques, such as Tomotherapy plans, could be used to potentially evaluate even better results for photon plans. In this study, the benefit from protons respect to photons was not clearly emerging regarding severe toxicities for cases in which optic chiasm, optic nerves and brainstem were strictly adjacent to the target volumes, in accordance with the above-mentioned literature. However, it is worth noting that the highest planning priority was given to the dose constraints for these OARs over target coverage in the optimization process. Consequently, NTCP values were small for both radiation techniques. Nevertheless, IMPT plans exhibit a clear advantage in terms of dose bath to the healthy tissues. This is due to the physical favorable properties of protons thanks to the inverse dose profile so allowing, in addition, the adoption of a limited number of field entrances. For this reason, the related intermediate toxicities predicted by the adopted NTCP models were expected to be less occurring. Thus, IMPT could be more advantageous when pursuing eye and brain function preservation strategies. In particular, conjunctivitis, keratitis and corneal ulceration could impact on QoL leading to the need of orbital ablation. Avoiding this dramatic sequela and preserving the organ without affecting outcome is mandatory in patients with good prognosis such as LA-PNSCs who had excellent response induction chemotherapy [45].

Regarding our selection criterion based on TS, the following further considerations should be mentioned. Firstly, the rational for the TS weighting factors in Equation (Equation 1) combines the clinical impact of the analyzed toxicities, the scientific breakthrough of the available NTCP models and some DVH parameters as pointed out in the quoted literature. Secondly, the cutoff value of 250 was arbitrarily chosen based on our clinical experience, taking particularly into account the QoL of patients affected by this kind of disease. A lower threshold would have led to select PT as a therapeutic choice, even in cases for which the dosimetric and NTCP model-based advantages would not have been of relevant clinical impact. In fact, when referring patients to PT, it is also necessary to consider the number of facilities currently in operation and the higher running costs, thus balancing the pro and cons.

Finally, for tumor localization involving the paranasal sinuses, the range uncertainty mainly due to air cavities filling stability during the radiation course, need to be adequately mitigated in PT. To this purpose, re-evaluation CT scans and possible replanning, in fact, are more than frequent for these patients. Therefore, when plan adaptation is needed, a dynamic scenario in which both the NTCP models and DVH parameters can vary during the treatment course must be considered.

## 5. Conclusions

For PNSC radiotherapy, dose-volume parameters alone may not sufficiently depict the relevance between VMAT and IMPT and the use of NCTP models comes to help in decision-making process. In definitive setting, the adoption of a mixed DVH-NTCP approach for comparison between VMAT and IMPT seems to be more advantageous in reduction of intermediate toxicities, thus potentially allowing organ preservation of the orbit.

## Figures and Tables

**Figure 1 cancers-14-02678-f001:**
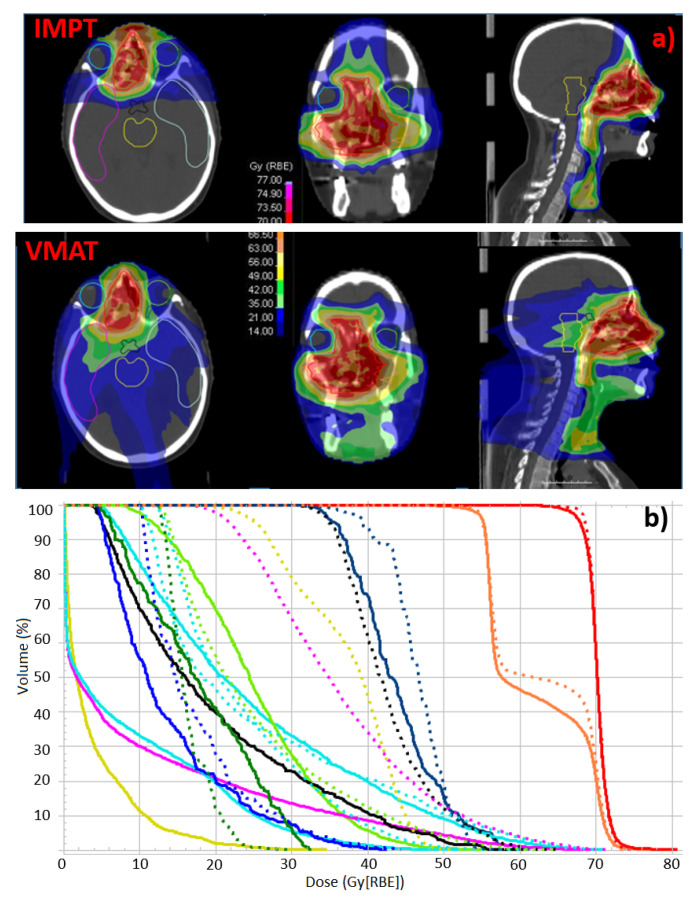
(**a**) Representative example of dose distribution for IMPT and VMAT plans. (**b**) DVH for the selected patient case and OARs; solid and dashed lines refer to IMPT and VMAT plans respectively.

**Table 1 cancers-14-02678-t001:** Dose planning goals are grouped in descending order of priority in column 2. D${}_{x}$ is the dose covering the x% of the structure volume while V${}_{x}$ is the volume of structure receiving x Gy(RBE). D${}_{mean}$ and D${}_{max}$ are the mean and the maximum dose, respectively. DVH parameters included in the study for plan comparison are listed in column 3. ALAP = As Low As Possible.

Structure	Planning Objectives/Constraints	DVH Indices
**Optic Chiasm**	D${}_{1\%}$ < 60 Gy(RBE)	V${}_{55}$
**Contralateral Optical Nerve**	D${}_{1\%}$ < 60 Gy(RBE), D${}_{10\%}$ ALAP	V${}_{55}$
**Brainstem**	D${}_{1\%}$ < 54 Gy(RBE)	
**Spinal Cord**	D${}_{1\%}$ < 45 Gy(RBE)	
**HR-CTV**	D${}_{95\%}$ > 95%	
	D${}_{2\%}$ < 107%	
	V${}_{95\%}$ > 95%	
**LR-CTV**	D${}_{95\%}$ > 95%	
	V${}_{95\%}$ > 95%	
**Ipsilateral Optical Nerve**	D${}_{1\%}$ < 60 Gy(RBE)	V${}_{55}$
**Ipsilateral Retina**	D${}_{mean}$ < 30 Gy(RBE); D${}_{max}$ < 60 Gy(RBE)	V${}_{50}$, V${}_{55}$
**Ipsilateral Eye**	D${}_{mean}$ < 30 Gy(RBE); D${}_{1cc}$ < 65 Gy(RBE)	
**Contralateral Retina**	D${}_{mean}$ < 30 Gy(RBE); D${}_{max}$ ALAP	
**Contralateral Eye**	D${}_{mean}$, D${}_{1cc}$ ALAP	
**Ipsilateral Ant. Chamber**	D${}_{mean}$ < 30 Gy(RBE)	
**Contralateral Ant. Chamber**	D${}_{mean}$ ALAP	
**Ipsilateral Lacrimal Gland**	D${}_{mean}$ < 30 Gy(RBE)	V${}_{30}$
**Contralateral Lacrimal Gland**	D${}_{mean}$ ALAP	
**Left Temp. Lobe**	D${}_{1cc}$ < 68 Gy(RBE)	V${}_{25}$, V${}_{35}$
**Right Temp. Lobe**	D${}_{1cc}$ < 68 Gy(RBE)	V${}_{25}$, V${}_{35}$
**Frontal Lobe**	D${}_{1cc}$ < 68 Gy(RBE)	V${}_{25}$, V${}_{35}$
**Brain minus LR-CTV**	D${}_{1cc}$ < 68 Gy(RBE)	V${}_{25}$, V${}_{35}$
**Ipsilateral Cochlea**	D${}_{mean}$ < 40 Gy(RBE)	
**Contralateral Cochlea**	D${}_{mean}$ ALAP	

**Table 2 cancers-14-02678-t002:** Details of the selected NTCP models for the listed organs/endpoints.

Toxicity Endpoint (Scoring)	Author	NTCP Model	OAR
Blindness (Late/Severe)	Burman et al. [12]	$NTCP=\frac{1}{2\pi }\int_{\infty }^{t}\exp^{\frac{-u^{2}}{2}}du,t=\frac{gEUD-TD_{50}}{m*TD_{50}}$	Optic Chiasm, Left/Right Optical Nerve
Brain Necrosis (Late/Severe)	Bender et al. [15]	$NTCP={\left(1+{\left(\frac{D_{50}}{EQD_{2}}\right)}^{4\gamma }\right)}^{-1}$	Brainstem, Brain outside CTV
Overall Ocular Toxicities (Acute/Intermediate)	Batth et al. [11]	$NTCP={\left(1+e^{-\beta_{0}-\beta_{1}*D_{max}}\right)}^{-1}$	Left/Right Lacrimal Gland
Temporal Lobe Necrosis (Late/Severe)	Kong et al. [35]	$NTCP={\left(1+e^{-\beta_{0}-\beta_{1}*D_{max}}\right)}^{-1}$	Left/Right/Frontal Lobe
Tinnitus (Late/Intermediate)	Lee et al. [37]	$NTCP=\frac{1}{2\pi }\int_{\infty }^{t}\exp^{\frac{-u^{2}}{2}}du,t=\frac{gEUD-TD_{50}}{m*TD_{50}}$	Left/Right Cochlea
Cataract Requiring Intervention (Late/Intermediate)	Burman et al. [12]	$NTCP=\frac{1}{2\pi }\int_{\infty }^{t}\exp^{\frac{-u^{2}}{2}}du,t=\frac{gEUD-TD_{50}}{m*TD_{50}}$	Left/Right Lens
Dry Eye Syndrome (Late/Severe)	Jeganathan et al. [38]	$NTCP=\frac{e^{\left(4\gamma \left(\frac{D}{D_{50}}-1\right)\right)}}{1+e^{\left(4\gamma \left(\frac{D}{D_{50}}-1\right)\right)}}$	Left/Right Lacrimnal Gland
Brain Necrosis (Late/Intermediate)	Niyazi et al. [16]	$NTCP={\left(1+\frac{39.5^{10}}{gEUD}\right)}^{-1}$	Brain oustide CTV

**Table 3 cancers-14-02678-t003:** Dosimetric comparison for target coverage between VMAT plans and IMPT plans. Data are expressed as mean values ± one standard deviation.

Target Volume	Dose Parameter	VMAT	IMPT
	D${}_{95\%}$	68.4 ± 0.7	67.4 ± 0.7
	D${}_{98\%}$	67.2 ± 1.4	65.9 ± 0.8
	V${}_{95\%}$	98.5 ± 1.4	97.1 ± 1.4
**HR-CTV**	D${}_{2\%}$	72.9 ± 0.6	73.4 ± 0.7
	D${}_{mean}$	70.4 ± 0.3	70.3 ± 0.4
	HI	0.08 ± 0.03	0.11 ± 0.02
	CI	1.64 ± 0.19	1.28 ± 0.28
	D${}_{95\%}$	55.0 ± 0.4	55.0 ± 0.4
	D${}_{98\%}$	54.5 ± 0.4	53.5 ± 1.0
**LR-CTV**	V${}_{95\%}$	99.6 ± 0.4	98.3 ± 0.4
	CI	1.45 ± 0.4	1.14 ± 0.3
	D${}_{mean}$	57.6 ± 2.3	57.0 ± 1.2

**Table 4 cancers-14-02678-t004:** Dosimetric analysis for OARs. Each cell contains the number of patient cases as a function of ΔDVH values: ΔDVH = +1 when IMPT performs better than VMAT, ΔDVH = −1 if VMAT performs better than IMPT and ΔDVH = 0 when no net benefit was found in either the two techniques.

ΔDVH for V${}_{30}$	1	0	−1	ΔDVH for V${}_{50}$	1	0	−1
Ipsilat. Lacrimal gland	6	1	1	Ipsilat. Retina	13	8	0
Contralat. Lacrimal gland	3	0	0	Contralat. Retina	11	7	1
**ΔDVH for V${}_{35}$**	**1**	**0**	**−1**	**ΔDVH for V${}_{25}$**	**1**	**0**	**−1**
Left Temporal Lobe	22	0	0	Left Temporal Lobe	22	0	0
Right Temporal Lobe	22	0	0	Right Temporal Lobe	22	0	0
Frontal Lobe	22	0	0	Frontal Lobe	22	0	0
**ΔDVH for V${}_{55}$**	**1**	**0**	**−1**	**ΔDVH for D${}_{mean}$**	**1**	**0**	**−1**
Optic Chiasm	11	10	1	Ipsilat. Lacrimal gland	6	16	0
Ipsilat. Optical Nerve	7	15	0	Contralat. Lacrimal gland	7	14	1
Contralat. Optical Nerve	12	8	0	Ipsilat. Anterior chamber	6	15	1
Ipsilat. Retina	11	7	1	Contralat. Anterior chamber	7	14	1
Contralat. Retina	10	6	1				

**Table 5 cancers-14-02678-t005:** Results for severe toxicities investigation. The number of patient cases with ΔNTCP${}^{s}$ higher than the selected thresholds is reported. Percentage values over the entire patient cohort is showed in brackets. Values in bold character fulfill the condition 1b.

OAR	0% ≤ ΔNTCP${}^{s}$ < 1%	1% ≤ ΔNTCP${}^{s}$ < 3%	ΔNTCP${}^{s}$≥ 3%
Contralateral Lacrimal Gland	22 (100%)	-	-
Ipsilateral Lacrimal Gland	18 (81.8%)	3 (13.6%)	**1 (4.5%)**
Optic chiasm	20 (90.9%)	2 (9.1%)	-
Contralateral optic nerve	22 (100%)	-	-
Ipsilateral optic nerve	22 (100%)	-	-
Left Temporal Lobe	10 (45.5%)	10 (45.5%)	**2 (9.1%)**
Right Temporal Lobe	11 (50.0%)	10 (45.5%)	**1 (4.5%)**
Frontal Lobe	17 (77.3%)	4 (18.2%)	**1 (4.5%)**
Brainstem	22 (100%)	-	-
Brain	22 (100%)	-	-

**Table 6 cancers-14-02678-t006:** Results for intermediate toxicities investigation. The number of patient cases with ΔNTCP${}^{i}$ values in the selected intervals is reported. Percentage values over the entire patient cohort is showed in brackets. Values in bold character fulfill the condition 1a.

OAR	ΔNTCP${}^{i}$ < −20%	−20%≤ΔNTCP${}^{i}$ < −5%	−5%≤ΔNTCP${}^{i}$ < 5%	5%≤ΔNTCP${}^{i}$ < 20%	ΔNTCP${}^{i}\geq 20\%$	ΔNTCP${}_{\mathrm{mean}}^{i}\left(\%\right)$
Brain	-	-	2 (9.1%)	12 (54.5%)	**8 (36.4%)**	17.3
Ipsilateral Lens	2 (9.1%)	2 (9.1%)	9 (40.9%)	5 (22.7%)	**4 (18.2%)**	3.4
Contralateral Lens	2 (9.1%)	3 (13.6%)	5 (22.7%)	5 (22.7%)	**7 (31.9%)**	10.0
Ipsilateral Lacrimal Gland	3 (13.6%)	4 (18.2%)	5 (22.7%)	7 (31.9%)	**3 (13.6%)**	1.4
Contralateral Lacrimal Gland	-	7 (31.9%)	6 (27.2%)	4 (18.2%)	**5 (22.7%)**	5.9
Ispilateral Cochlea	-	-	4 (18.2%)	13 (59.1%)	5 **(22.7%)**	12.5
Contralateral Cochlea	-	-	5 (22.7%)	10 (45.4%)	7 **(31.9%)**	16.1

**Table 7 cancers-14-02678-t007:** Results for selection criteria evaluation. Values in bold character indicate patient cases fulfilling the selected condition criteria. Columns 4–6 show results for each single term in Equation (Equation 1), while the TS values are listed in the last column.

Patient	ΔNTCP${}^{i}$ > 20%	ΔNTCP${}^{s}$ > 3%	w${}_{1}\Sigma \Delta {\mathrm{NTCP}}^{s}$	w${}_{2}\Sigma \Delta {\mathrm{NTCP}}^{i}$	w${}_{3}\Sigma \Delta \mathrm{DVH}$	TS > 250
**P1**			200	65	110	**375**
**P2**		**DES**	280	72	110	**462**
**P3**			60	68	120	248
**P4**			100	130	140	**370**
**P5**			100	91	120	**311**
**P6**			100	121	80	**301**
**P7**		**Brain Necrosis**	140	66	130	**336**
**P8**			60	78	90	228
**P9**	**G2 Brain Necrosis + Tinnitus + Catharact**		160	51	120	**331**
**P10**			100	38	120	**258**
**P11**		**Brain Necrosis**	180	19	120	**319**
**P12**			120	82	40	242
**P13**	**G2 Brain Necrosis + Tinnitus + Catharact**		160	82	100	**342**
**P14**			80	75	100	**255**
**P15**			140	-5	90	225
**P16**	**G2 Brain Necrosis + Tinnitus + Ocular tox**		140	93	120	**353**
**P17**			80	38	140	**258**
**P18**			120	-8	60	172
**P19**		**Brain Necrosis**	120	-6	70	184
**P20**			120	75	140	**335**
**P21**			100	98	130	**328**
**P22**	**G2 Brain Necrosis + Tinnitus + Ocular tox**		140	145	110	**395**

## Data Availability

The data presented in this study are available on request from the corresponding author.

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
