# Peer review of "A Patient Selection Approach Based on NTCP Models and DVH Parameters for Definitive Proton Therapy in Locally Advanced Sinonasal Cancer Patients"

_cancers, 2022, doi:10.3390/cancers14112678_

Round 1
Reviewer 1 Report
The article reports a model-based approach to select patients for radiotherapy with protons in locally advanced sinonasal cancer. The analysis was based on various NTCP models. Although, as was discussed by the authors, further considerations should be considered for practical application, the methodology presented in this paper is of considerable interest. I think that following comments should be considered.
Comments:
1) Describe in more detail how Wilcoxon singled-rank test was applied to calculate the third term of formula (1). Delta_DVH was firstly defined as 100*[(phDVH-pDVH)/phDVH]. But Delta_DVH was set to be -1 or 0 or 1 in accordance with Delta_DVH. I was confused. What is Delta_NTCP^W_r in the third term of formula (1)? Is it Delta_DVH^W_r? How much is m?
2) In this study, three horizontal fixed beams were employed for proton therapy. However, proton therapy is generally conducted by using gantry with rotating couch. The authors should discuss it in terms of achievable dose distributions.
Reviewer 2 Report
The manuscript presents a comparative planning study and suggests a selection criterion of SNC patients for IMPT or VMAT. It is very well written and I really enjoyed the opportunity to read it.
Please find below a detailed list of comments.
Introduction:
- 2 l. 37-39: Why a constant RBE = 1.1 (to both tumor and healthy tissues) should be considered an advantage? I suggest removing the sentence.
- 2 l. 64-66: The concept seems not clear. In particular, does “new” refers to “recent” NTCP models (built for only photon therapy, for instance), or to models that are yet to be trained and validated for both photons and protons? Please clarify.
- 2 l. 68-69 (and later in the study design): I agree with the premise that NTCP models built on photon cohorts are not guaranteed to work properly on protons. Hence, I do not understand how the authors solve this issue before performing the plan comparison analysis. This contradiction appears at least equally striking for the comparison of DVH metrics supposed to be associated to certain toxicity outcomes.
Materials and Methods:
- 4 l. 154-155 (and p. 5 l. 173): I believe that it would be interesting to assess also the PTV coverage, irrespective of how the loss function for the optimization was defined. In any case, it cannot be stated that the definition of the objective function is used in this way as a criterion for the choice of the way to assess the results.
- 5 l. 173-174: I do not understand the meaning of the premise “For the sake of clarity and easiness and to avoid case-specific bias”. Might it be simply dropped? If not, please clarify.
- 5 l. 178-182: First, a formal concern. It seems that the authors adopted a computer science formalism by redefining the variable ∆DVH (similarly to customary code lines like x = x + 1), which however might be unusual from a mathematical point of view. Accordingly, I would suggest to introduce a new symbol for the categorical function of the continuous ∆DVH. Second, I believe that setting a unique threshold for the percentage change of any DVH metrics – independently from the severity of the associated risk – can be misleading. Essentially, why the evaluation of ∆DVHs in TS (p. 6, eq. 1) was not split in two different sums, weighted according to the toxicity severity, as it has been done for the ∆NTCPs? Third, could the authors comment on the choice to categorize a set of continuous variables (with the inherent loss of information)?
- 6 eq. 1: I am not a clinician, so I just ask the authors to clarify (not necessarily in the manuscript) a point: does any of the considered DVH metrics impact on the toxicities evaluated through the NTCP models considered later on? If yes, one might argue that some toxicities have been considered more than once in the TS expression.
- 6 l. 217-223: why do the authors just focus on one side of the therapeutic window (they correctly specify “in terms of reduced risk of radiation-induced side effects”), without suggesting a generalized version of TS, which could include also the evaluated CI and HI? From my point of view, this issue (possibly along with the problem of scope of NTCP and DVH metrics discussed for p. 2 l. 68-69) is the major limitation of the study.
Minor comments:
- 6 eq. 1: Please substitute ∆NTCPrW with ∆DVHrW. There might be a residual note at the end of p. 6 l. 213.
p.6 l. 228: Please specify the Figure number.
Round 2
Reviewer 1 Report
I am satisfied with the revisions that have been made by the authors.
Reviewer 2 Report
The authors addressed my issues.